# How to Estimate Optimal Malaria Readiness Indicators at Health-District Level: Findings from the Burkina Faso Service Availability and Readiness Assessment (SARA) Data

**DOI:** 10.3390/ijerph17113923

**Published:** 2020-06-01

**Authors:** Toussaint Rouamba, Sekou Samadoulougou, Cheick Saïd Compaoré, Halidou Tinto, Jean Gaudart, Fati Kirakoya-Samadoulougou

**Affiliations:** 1Centre de Recherche en Epidémiologie, Biostatistique et recherche clinique, Ecole de Santé Publique, Université Libre de Bruxelles (ULB), Route de Lennik, 808 B-1070 Bruxelles, Belgien; Toussaint.Rouamba@ulb.ac.be; 2Clinical Research Unit of Nanoro, Institute for Research in Health Sciences, National Center for Scientific and Technological Research, 42, Avenue Kumda-Yoore, BP 218 Ouagadougou CMS 11, Ouagadougou, Burkina Faso; halidoutinto@gmail.com; 3Evaluation Platform on Obesity Prevention, Quebec Heart and Lung Institute Research Center, Quebec City, QC G1V 4G5, Canada; ouindpanga-sekou.samadoulougou.1@ulaval.ca; 4Centre for Research on Planning and Development (CRAD), Laval University, Quebec City, QC G1V 0A6, Canada; 5National Malaria Control Programme, Ministry of Health, 03 BP 7009 Ouagadougou, Burkina Faso; cheick.said@gmail.com; 6Aix Marseille Univ, APHM, INSERM, IRD, SESSTIM, Hop Timone, BioSTIC, F-13005 Marseille, France; jean.gaudart@univ-amu.fr; 7Aix Marseille Univ, IRD, INSERM, UMR1252 Sciences Economiques & Sociales de la Santé & Traitement de l’Information Médicale, 13385 Marseille, France

**Keywords:** SARA survey, binomial hierarchical Bayesian, geo-epidemiology, spatial analysis, malaria, service readiness, health district, Burkina Faso

## Abstract

One of the major contributors of malaria-related deaths in Sub-Saharan African countries is the limited accessibility to quality care. In these countries, malaria control activities are implemented at the health-district level (operational entity of the national health system), while malaria readiness indicators are regionally representative. This study provides an approach for estimating health district-level malaria readiness indicators from survey data designed to provide regionally representative estimates. A binomial-hierarchical Bayesian spatial prediction method was applied to Burkina Faso Service Availability and Readiness Assessment (SARA) survey data to provide estimates of essential equipment availability and readiness for malaria care. Predicted values of each indicator were adjusted by the type of health facility, location, and population density. Then, a health district composite readiness profile was built via hierarchical ascendant classification. All surveyed health-facilities were mandated by the Ministry of Health to manage malaria cases. The spatial distribution of essential equipment and malaria readiness was heterogeneous. Around 62.9% of health districts had a high level of readiness to provide malaria care and prevention during pregnancy. Low-performance scores for managing malaria cases were found in big cities. Health districts with low coverage for both first-line antimalarial drugs and rapid diagnostic tests were Baskuy, Bogodogo, Boulmiougou, Nongr-Massoum, Sig-Nonghin, Dafra, and Do. We provide health district estimates and reveal gaps in basic equipment and malaria management resources in some districts that need to be filled. By providing local-scale estimates, this approach could be replicated for other types of indicators to inform decision makers and health program managers and to identify priority areas.

## 1. Introduction

Since the early 2000s across Sub-Saharan African (SSA) countries, the burden of malaria has considerably declined. This is partly explained by a major increase in the mobilization of funding and scaling up of malaria control interventions [1,2,3]. Although malaria-related deaths have significantly decreased worldwide, the 2018 World Health Organization (WHO) report stated that all WHO regions reported either only slight progress or an increase in the malaria incidence rate [3].

Unfortunately, seven of the SSA countries, including Burkina Faso, account for more than half of all malaria cases and deaths worldwide [3]. Despite reasonable or even good coverage for most interventions, the annual malaria incidence remains stubbornly high, having disproportionate effects on the health of young children and pregnant women [4]. Reliable health services, including health services’ availability and geographical, sociocultural, and financial accessibility, are essential for effectively improving health outcomes, especially malaria incidence cases and case fatality rates [5,6,7]. The WHO and its partners proposed a general framework to help SSA countries to monitor and assess periodically their health system performance called the Service Availability and Readiness Assessment (SARA) [8]. The SARA survey generates a set of tracer indicators of service availability and readiness that can be used alongside other indicators to support national and local administrators with planning and managing health systems, including adequate allocation of health services, human resources, and availability of medicines and supplies [8]. SARA survey data provide crucial information to ensure that health facilities are resourced and equipped to deliver essential care to the population. However, most health programs to prevent or reduce diseases are implemented at the health-district level (operational unit), whereas the SARA survey provides service-specific readiness estimates with exclusive focus on national and regional rates.

In Burkina Faso, like other SSA countries, the new Global Technical Strategy for Malaria 2016–2030 approach involves transforming malaria surveillance through the District Health Information System (DHIS) into a core intervention [9]. The WHO further recommends malaria surveillance through the DHIS to be integrated into national and local (health district) malaria control strategies. Therefore, understanding the availability and readiness of health systems at the local scale (health district, community, or village) is necessary to assess whether health programs are progressing as planned or whether adjustments are needed at the operational unit of the health system [10]. Several field studies suggested that obtaining information about the performance of health programs as well as the disease burden at the local level might provide the information required to implement a highly efficient integrated approach to control disease transmission [11,12,13,14,15]. Unfortunately, data on the availability and readiness of health systems and malaria control programs are regionally representative [16]. Given the indicator variability within a region (between health districts), this information provides a limited overview at the district level. Several potential pitfalls exist when using nationally or regionally representative data to estimate indicators at the health-district level [17,18]. One general problem is the representativeness of the study population due to the small sample size. The other challenge is related to the pseudo-replication of information (autocorrelation) as well as microvariation within the same region.

Since conducting national censuses (or complete spatial coverage) is not always easy and obtaining a large sample size is not always feasible, finding innovative methods that can provide unbiased estimates at the health-district level is crucial. A current development in advanced statistical methodology, hierarchical spatial modeling [19,20,21,22,23,24,25] implemented in a Bayesian framework [19,26,27,28,29,30], provides opportunities to overcome these limits by providing reliable representative estimates at the local scale for improved data and decision making. A hierarchical Bayesian spatial modeling approach can be used to handle missing data, such as health districts not covered by the survey, using unknown parameters during estimation. So, the random variable values describing a site can be predicted by the neighboring sites’ data.

To address the growing need for and interest in subnational statistics, the purpose of this study was to estimate malaria readiness indicators at the health-district level from survey data designed to be regionally representative.

## 2. Methods

### 2.1. Study Setting

A survey was conducted in Burkina Faso, which is a landlocked country with a surface area of 274,200 km^2^ and a population of 20.2 million in 2018, of which 77.3% reside in rural areas [31]. The country is subdivided into 13 regions, 45 provinces, 351 communities, and 9000 villages. Burkina Faso’s climate is tropical and Sudanese in nature, with alternating rainy (from July to October) and dry seasons. The country’s epidemiological profile is marked by a high morbidity of endemic-epidemic diseases and a progressive increase in noncommunicable diseases. The main diseases of public health importance include malaria, acute respiratory infections, malnutrition, HIV/AIDS, tuberculosis, and sexually transmitted infections [4]. Burkina’s health system has three levels: Central, intermediate, and peripheral. The peripheral level is composed of 70 health districts, which form the operational entity of the national health system. Health care is provided by public and private institutions. The main obstacle to accessing health care is affordability; in 2016, to address this issue, the Burkinabe government initiated a subsidy program that provides free health services to children under 5 and pregnant women. Since 2012, the Ministry of Health has administered the SARA survey every two years to assess and monitor the availability and readiness of health facilities to provide quality health services.

### 2.2. Study Design, Sample, and Sampling Procedure

The study was designed as a health facility-based cross-sectional survey conducted between October and November 2014. This survey included both public and private health facilities across the three levels of health system organization (central, intermediate, and peripheral) and different locations (rural or urban). The sampling procedure used in this survey was reported previously [16]. Briefly, the study used stratified sampling with simple random sampling applied within each stratum, so that the indicators were representative at the regional level. For our study, the analysis included data collected at the health-district level. A total of 753 health facilities located in 70 health districts in the 13 regions of the country were included in the survey, representing 37.3% of all health facilities in Burkina Faso. All these health facilities provided malaria diagnosis and treatment services [16].

### 2.3. Data Collection and Processing

In this analysis, we assessed (1) the general operational capacity of services, with a focus on the availability of essential equipment, and (2) the readiness to provide malaria case management. The data were collected using two methods: Face-to-face interviews with the heads of health facilities or any other relevant health personnel, and direct observation to verify the availability, functionality, and use of the key items.

The assessment of the availability of essential equipment included weighing scale for children and adults, medical thermometer, stethoscope, tensiometer, latex gloves for physical exam, and a light source. To assess the service availability for malaria management, the following items were included: Malaria diagnosis by clinical symptoms, malaria diagnosis by rapid diagnostic test (RDT), malaria diagnosis by microscopy, malaria treatment, intermittent preventive treatment during pregnancy (IPTp), national guidelines for malaria treatment and IPTp, first-line antimalarials in stock, and RDT availability. In the analysis, the variable "health facility provides malaria diagnosis and treatment services" was not included as all (100%) health facilities surveyed provided malaria diagnosis and treatment services.

In brief, the availability and readiness indicators were binary variables, taking a value of 1 if the key item was available and in a functional state at the health facility, and 0 otherwise [8,16]. Data were recorded on a paper questionnaire. After verification and validation, data were entered electronically in a database designed as a Census and Survey Processing System (CSPRO).

### 2.4. Current Statistical Analyses Applied to SARA Survey Data in Burkina Faso

According to WHO recommendations, both tracer indicators, general and specific indices were used in routine data analysis [8,16,32]. A descriptive analysis was applied to the data to provide regionally and nationally representative estimates; this allowed the percentage of facilities providing specific services with tracer items or owning the equipment on the day of the assessment to be estimated. Beyond these descriptive statistics, health facility readiness indicators are also increasingly being used in SSA countries to assess the health system strengthening through the construction of a composite score [33,34,35]. In some studies, either principal component analysis (PCA) [36,37,38,39] or multiple correspondences analysis (MCA) [40,41] was directly applied to readiness indicators, which are usually defined as binary variables [8,16,40].

### 2.5. Our Analytical Approach

#### 2.5.1. Hierarchical Bayesian Spatial Modeling (HBSM)

In this study, four steps for modeling the SARA survey data were used to create a HBSM framework model for the data.

• Step 1: Model for the Data

As mentioned above, the availability or readiness variables were binary (denoted Y), each with its own Boolean-valued outcome, i.e., success (Y is equal to 1 with probability *p*) or failure (Y is equal to 0 with probability *q* = 1 − *p*). The Burkina Faso area comprises a set of K=1,…,70 non-overlapping health districts $S=\left\{S_{1},\ldots \ldots \ldots ,S_{K}\right\}$, and availability or readiness indicators are recorded for each health district. Let $n_{k}$ be the number of health facilities in the health district K ($S_{k}$). In each $S_{k},$ the variable Y is tested (or measured) $n_{k}$ times, and the number of successful trials among $n_{k}$ tests is counted. The probability of observing exactly $y_{k}$ successful trials for the variable Y among $n_{k}$ trials in the health district K, is:(1)$$p\left(y_{k}/n_{k},p_{k}\right)=\left(\begin{array}{c} n_{k}\\ y_{k} \end{array}\right)p_{k}{}^{y_{k}}{\left(1-p_{k}\right)}^{n_{k}-y_{k}}$$

• Step 2: Non-Spatial Grouped Binomial Regression (Proportional Counts) and Test for Spatial Autocorrelation

Since the analysis assumes that the variable Y in health district K follows a binomial distribution, the following model can be fitted to the data:(2)$$p\left(y_{k}/n_{k},p_{k}\right)\;\sim \;Binom\left(n_{k},\;p_{k}\right)$$
(3)$$logit\left(p_{k}\right)=log\left(\frac{p_{k}}{1-p_{k}}\right)=X_{k}^{T}\beta$$
where $p_{k}$ is the probability (proportion) of success in health district *k* ($S_{k}$) after considering the observed effects for covariates $X_{k}$ and β is the regression coefficients of the covariates. The $logit\left(p_{k}\right)$ is used in this general linear model to fit the probability of success $p_{k}$ as a linear combination of observed characteristics $X_{k}$.

In this study, a grouped binomial model was fitted for the number of successes $y_{k}$ among $n_{k}$ trials with probability $p_{k}$ after considering covariate effects, including the type (number of private health facilities), location (number of rural health facilities), and population density. Then, spatial autocorrelation was quantified in the residuals of the model with Moran’s *I* statistic and a permutation test.

• Step 3: Model for Spatial Random Effects and Prior Distribution of the Model Parameters

In this step, we fit the spatial dependence in the data by including spatial random effects in the model. In this model, health district random effects are included, and region identities are included as random factors to account for interregional variance not captured by the fixed effects and health district specific random effects:(4)$$logit\left(p_{k}\right)=log\left(\frac{p_{k}}{1-p_{k}}\right)=X_{k}^{T}\beta +\psi_{k}+z_{R}$$
where $p_{k}$ represents probabilities that are assumed to have beta prior distributions, $p_{k}\;\sim \;\mathrm{Beta}\left(\alpha_{1}=1,\;\alpha_{2}=1\right)$, $\beta =\left(\beta_{0},\;\beta_{1},\beta_{2},\beta_{3}\right)\;\sim \;Normal\left(0,\;\tau_{\beta }\right)$ is the unstructured fixed effects, $\psi_{k}=(\psi_{1},\ldots ,\psi_{70})$ represents the health district spatial random effects (i.e., residual area variation arising from unmeasured or unknown factors), and $z_{R}=\left(z_{1},\;\ldots ,\;z_{13}\right)\sim Normal\left(0,\;\tau_{z}^{2}\right)$ represents regional area random effects.$\;\mathrm{The}\;\tau_{z}^{2}$ is the variances of the marginal regional area random effect. The $\psi_{k}$ is decomposed as the sum of a structured spatial random effect ($u_{k}$) and an unstructured random effect ($\nu_{k}$) [42]. A neighborhood structure to control for the spatial effect between health districts was included [43]. This neighborhood structure is a binary adjacency weight (W) based on border sharing. Two health districts, $S_{k}$ and $S_{j}$, are neighbors if they share a common boundary, whereby $w_{kj}$ is assigned 1 when $S_{k}$ and $S_{j\neq k}$ share a common border. If no common border exists, 0 is assigned. In this study, Besag–York–Mollié (BYM) spatial structure, which assumes the presence of two underlying spatial patterns, was used to model the spatial autocorrelation [42]:(5)$$\psi_{k}=u_{k}+\nu_{k}$$

A conditional autoregressive (CAR) prior distribution and Gaussian prior distribution was used to fit $u_{k}$ and $\nu_{k}$, respectively.
(6)$$u_{k}|u_{j\neq k},\;W\;\sim \;Ν\left(\frac{\sum_{i=1}^{k}w_{kj}u_{j}}{\sum_{i=1}^{k}w_{kj}},\;\frac{\tau_{u}^{2}}{\sum_{i=1}^{k}w_{kj}}\right)$$
(7)$$\nu_{k}\sim Normal\left(0,\;\tau_{v}^{2}\right)$$
where $\overset{k}{\underset{i=1}{\sum }}w_{kj}$ denotes the number of neighbors for health district $S_{k}$ and $\tau_{u}^{2}$ and $\tau_{v}^{2}$ are the variances of the marginal structured and unstructured components, respectively.

• Step 4: Hyperparameter Specification for Prior Distribution of the Model

This step consists of fixing the prior distributions for each of the unknown hyperparameters of the model ($\tau_{\beta }$, $\tau_{u}$, $\tau_{v}$, and $\tau_{z}$). Since no prior knowledge is available regarding the model hyperparameters, they may be assumed to be independent and have vague or minimally informative hyperpriors. This assumption allows the data to play the main role in determining posterior distributions. All sets of random effects are zero-mean centered. Non-informative prior distribution $\beta \sim \;Normal\left(0,\;\tau_{\beta }\right)$ is used for the coefficients, with $\tau_{\beta }=100$. However, the specification uses minimally informative priors on the log of the structured effect precision, $\log\left(\tau_{u}\right)\sim \;logGamma\left(1,\;0.0005\right),$ and the unstructured effect precision is $\log(\tau_{u})\sim \;logGamma\left(1,\;0.0005\right)$. Non-informative prior distribution $\tau_{z}\;\sim \;Uniform\left(0,\;100\right)$ is used for regional spatial random effect precision.

#### 2.5.2. Bayesian Implementation and Goodness of Fit

In this study, to obtain the posterior marginal distribution of model parameters and fitted values, a hierarchical Bayesian model was fitted. The Bayesian computing was performed using the integrated nested Laplace approximation (INLA), which is a validated, reliable, and effective alternative to the Markov chain Monte Carlo (MCMC) method [44,45,46]. Each model fitted from SARA indicators was adjusted by the type of health facility, the location, and the population density. The median and the 2.5% and 97.5% quantiles of the fitted values were used as the health-district level estimates, with a 95% credible interval (95% CrI). To facilitate the interpretation of the results, the predicted posterior means of fitted values for each indicator were categorized into quartiles and then mapped.

For the assessment of the goodness of fitting, we firstly executed the different models without the adjustment variables and secondly with the adjustment variables. We then checked whether there was an improvement of the deviance information criterion (DIC). The internal validation (predictive performance) of the models was assessed using a leave-one-out cross-validation score, the conditional predictive ordinate (CPO) approach [47]. The convergence of the model parameters was assessed graphically.

#### 2.5.3. Composite Readiness Profile Building through Hierarchical Ascendant Classification

We performed a Hierarchical Ascendant Clustering (HAC) on the predicted values of availability and readiness to assess the resemblances and differences between health districts from a multidimensional point of view. For the HAC, Euclidean distance and Ward’s criterion were used. Ward’s criterion is based on the Huygens–Steiner theorem, which allows decomposition of the total inertia between and within group variance. A group or cluster is an aggregation of several similar health districts. In the initial step of the algorithm, all clusters are singletons (clusters containing a single point). Ward’s approach consists of aggregating two groups so that the growth within-inertia is minimal in each step of the algorithm. This method minimizes the total within-cluster variance and maximizes the total intercluster variance. To simplify the use of the study findings by health system administrators, the final result groups all health districts into clusters or composite readiness profiles.

## 3. Results

### 3.1. Repartition of Sampled Health Facilities

Table 1 shows the repartition of sample size according to region, location, and type of governing authority. About two-thirds of the health facilities were in rural areas and four-fifths were public facilities.

### 3.2. Availability Scores for Essential Equipment

The predicted values of health district essential equipment rates from HBSM are shown in Figure 1 and Appendix A. The estimated rate for adult weighing scale availability was fairly consistent across regions (92.0% to 96.9%). In contrast, the infant weighing scale availability varied widely (40% to 94.3%). More than 75% of the health facilities in each health district had high scores for stethoscopes (98.7%), medical thermometers (99.7%), and blood pressure apparatus (97.3%). The light source availability varied widely across the country. The rate was lowest in the health districts located in the Boucle de Mouhoun and Cascade regions (between 22.2% and 58.7%), and highest in the health districts located in the central region (>79.6%) as well as in health districts of Dafra and Dori, among others.

### 3.3. Malaria Readiness Scores

The results showed an interregional variability for malaria readiness indicators throughout the study area, albeit this variation was low (Appendix A). Figure 2, Figure 3, Figure 4 and Appendix A summarize the geographical distribution of malaria readiness at the health-district level.

The geographical distribution of malaria readiness indicators regarding malaria diagnostic at the health-district level is presented on Figure 2. The RDT availability rate on the day of the survey ranged from 44.9% to 94.3%. More than three-quarters of districts require clinical signs for malaria diagnosis (>98%). Overall, health districts located in the political capital, Ouagadougou, and the economic capital, Bobo-Dioulasso, had the lowest rates for the use of rapid diagnostic tests. However, in these regions, the level of microscopy use for malaria diagnostic was high.

The Figure 3 shows the geographical distribution of malaria readiness indicators regarding malaria treatment at the health-district level. We observed a heterogeneity in antimalarial drug availability across the country. For the first-line antimalarial drugs (artemisinin-based combination therapy, ACT for short), 25% of health districts had a score less than 94.3%, whereas about three-quarters of health districts had a percentage less than 46.5% in terms of disposal of artesunate parenteral form. More than three-quarters of districts had a high rate (>91.2%) in terms of providing IPTp. Health districts located in major cities, such as Ouagadougou and Bobo-Dioulasso, had the lowest rates regarding the ACT in stock. The percentage of health districts with no ACT in stock on the day of survey ranged from 2.4% to 12.4%.

Health districts located in Ouagadougou and Bobo-Dioulasso had the lowest rates of staff training on malaria diagnosis and treatment and IPTp (Figure 4).

### 3.4. Composite Readiness Profile for Malaria Case Management

Figure 5 displays the geographical distribution of health district composite readiness profiles for malaria case management and Table 2 summarizes the characteristics of each profile. According to the HAC results, three malaria composite readiness profiles were established among the 70 health districts. A total of 44 (62.9%), 19 (27.1%), and 7 (10%) health districts were classified as high, medium, and low readiness, respectively.

Compared with health districts with low and medium readiness profiles, the high composite readiness performance profile is characterized by a high rate of availability of RDTs, a high rate of first-line antimalarial drugs, including rectal or injectable forms, and a high rate of availability of IPTp. Spatially, the low performance was found in urban areas in the central and Haut Bassins regions. The variables (tracer indicators) that best characterized each composite readiness profile are summarized in Appendix A.

## 4. Discussion

To date, the Service Availability and Readiness Assessment (SARA) surveys were carried out in Burkina Faso to generate estimates on basic equipment availability and the readiness to provide specific health care (such malaria case management) for the whole population including pregnant women, mothers, and children at regional level (with smaller sample size). This paper provides an alternative method based on HBSM for optimal estimation of subnational indicators drawn from health facility-based survey data with a much smaller sample size. Since local administrators need information on the operational scale for planning purposes, this alternative method, using advanced statistical methods applied to SARA survey data, offers a useful method for countries with limited resources. This method has been used in other areas to estimate indicators at the subnational level from samples drawn for national or regional estimates [17,18].

The results of our modeling are valuable for informing malaria control program managers, technical and financial partners, and Ministry of Health officials at the health district, provincial, regional, and central levels. The findings regarding the tracers of the readiness for services in the management of malaria indicated deficiencies that vary between health districts and need to be addressed by all stakeholders.

Through the application of the Bayesian method, the problem of sample size is minimized [26,27,28,29,30] as Bayesian approaches are not asymptotic-based, which is a feature that can be an obstacle to the use of frequentist methods in small sample contexts. Consideration of the spatial autocorrelation between health districts provides a more reliable estimate. According to this statistical principle, an event in the neighborhood closest to another may not necessarily increase the information available in the data if similar to the one already assessed [48]. Consequently, such assessments only increase the sample size without providing a complete set of information that is independent [49].

This study highlighted the gaps that must be addressed to improve the quality of health facility-based malaria cases management. The study showed the low rates of basic equipment throughout the study area, especially for two elements: Infant weighing scales and light sources. Without an infant weighing scale, health facility staff are often forced to prescribe drugs based on age when determining the optimal dosage of antimalarial drugs. This can lead to under- or over-dosing of drug prescriptions. Local health authorities should strengthen the availability of this equipment in health facilities located in the north, Cascades, and south-central regions. Health districts with high rates of light source availability were mostly located in urban areas, where electricity is more readily available. In rural areas, until the government finds the means to provide electricity, renewable energy sources, such as solar panels, could be an alternative to providing energy for light sources.

Notably, the current policy of routine parasitological diagnosis of malaria in Burkina Faso is based on the use of RDTs [50]. Although approximately 75% of health districts have an RDT coverage of more than 94.2%, this analysis indicated that RDT availability is not optimal across the country. For health districts located in urban regions, the RDT coverage was low. This suggests that patients continue to be misdiagnosed for malaria and mistreated. The results also showed that health districts located in urban areas had low rates of other malaria readiness indicators, especially the availability of antimalarial drugs. This finding implies that patients in these areas still use other sources of drug supplies, such as private pharmacies, drug stores, and/or private medical centers, where patients can be diagnosed and purchase over-the-counter medications. The disparity in malaria readiness coverage, especially the inadequate coverage of RDTs and ACTs in urban regions, could hinder the effectiveness of the National Malaria Control Program (NMCP), the “test, treat, track” strategy; therefore, the NMCP must seek to identify gaps and optimize resource distribution in health districts with low coverage. Given that in the Burkina Faso context the evaluation of SARA indicators is assessed every two years and given that our results found that there is a variability between health districts, it might be interesting that health authorities increase the frequency of SARA surveys, especially for health district with weak performances; for instance, every six months until a desired threshold has been reached and every one after the threshold is reached, and then 2 years.

A limitation of the study was that a cross-sectional survey was used as design, so the availability of basic equipment and malaria readiness service may vary over time. Another limitation is the fact that estimation of our indicators at the subnational level was based on statistical models, which may not really reflect the real situation. For the upcoming SARA surveys, we suggest a nested study where a region would be selected, in which a representative sample at the health-district level will be performed to serve as a control district for the validation of our model.

## 5. Conclusions

Our study provides estimates at the health-district level using existing data designed to be regionally representative. We showed that HBSM could be a useful tool to enable the use of regionally representative data with a small sample size to estimate rates (with uncertainty) of malaria readiness indicators at the health-district level. The results indicated gaps in basic equipment availability and resources in some health districts, which must be addressed. In a limited resource setting, health programs may struggle to operate effectively due to the lack of reliable estimates at the operational level for monitoring purposes. As demonstrated here, our proposed approach could be replicated for other types of indicators to provide local level estimates for local policy makers so that the gaps can be targeted and addressed as a priority. Our results suggest that further investigations should be implemented to assess the impact of the Health district composite readiness score on the spatial distribution of malaria burden.

## Figures and Tables

**Figure 1 ijerph-17-03923-f001:**
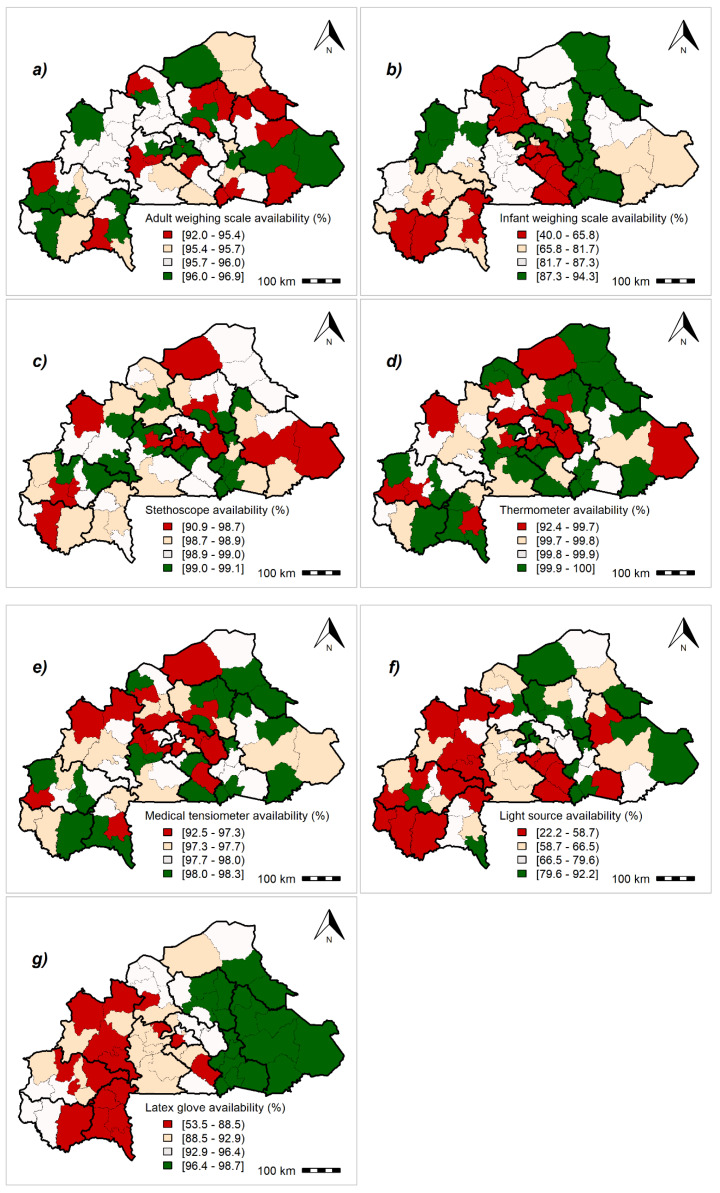
Geographical distribution of the availability of essential equipment at the health-district level: Posterior means of fitted values. (**a**) Adult weighing scale, (**b**) infant weighing scale, (**c**) stethoscope, (**d**) thermometer, (**e**) blood pressure apparatus, and (**f**) light source and (**g**) latex glove. Maps created by Toussaint Rouamba et al., 2019. Source of materials: The shapefile was obtained from the “Base Nationale de Découpage du territoire” of Burkina Faso (BNDT, 2006). The Service Availability and Readiness Assessment data for modelling were obtained from the Ministry of Health of Burkina Faso.

**Figure 2 ijerph-17-03923-f002:**
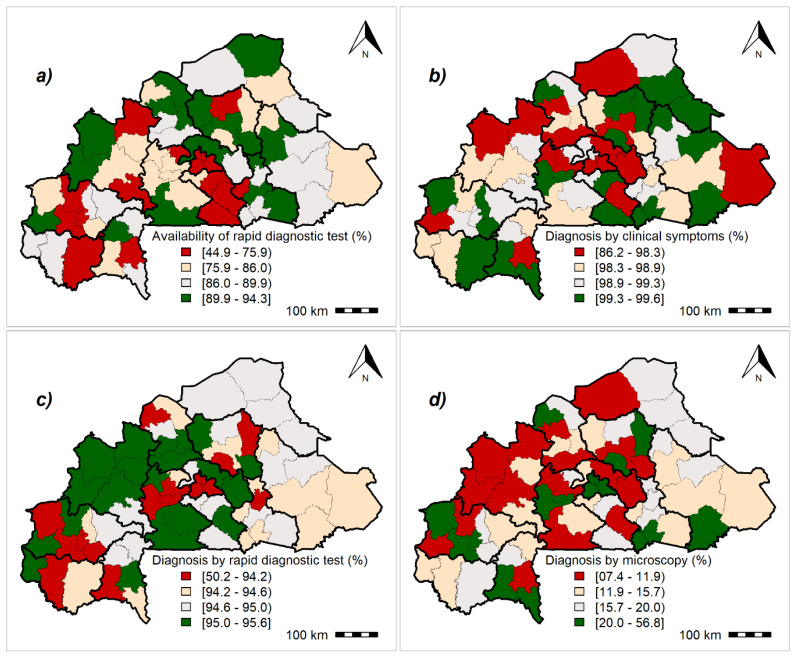
Geographical distribution of malaria readiness indicators regarding malaria diagnostic at the health-district level: Posterior medians of fitted values. (**a**) Availability of malaria rapid diagnostic test, (**b**) malaria diagnosis by clinical symptoms coupled with parasitological diagnosis, (**c**) malaria diagnosis by rapid diagnostic test, (**d**) malaria diagnosis by microscopy. Maps created by Toussaint Rouamba et al., 2019. Source of materials: The shapefile was obtained from the “Base Nationale de Découpage du territoire” of Burkina Faso (BNDT, 2006). The Service Availability and Readiness Assessment data for modelling were obtained from the Ministry of Health of Burkina Faso.

**Figure 3 ijerph-17-03923-f003:**
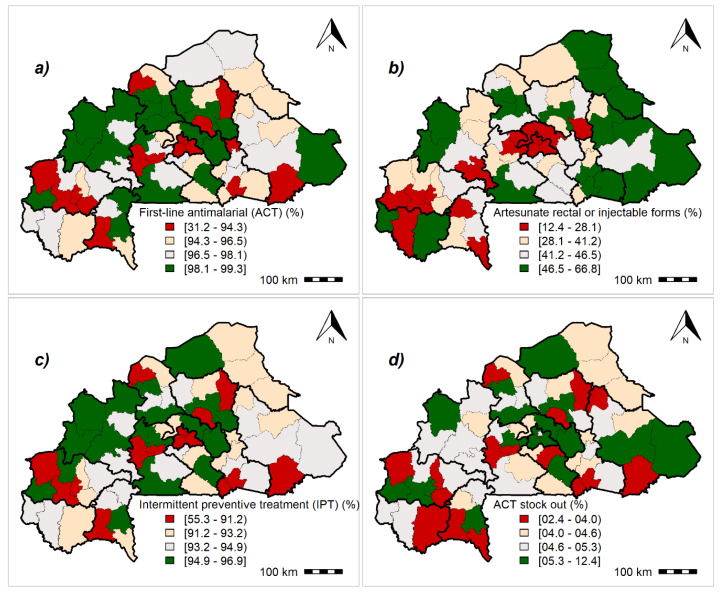
Geographical distribution of malaria readiness indicators regarding malaria treatment at the health-district level: Posterior medians of fitted values. (**a**) First-line antimalarial (artemisinin-based combination therapy), (**b**) artesunate rectal or injectable forms, (**c**) intermittent preventive treatment during pregnancy, (**d**) artemisinin-based combination therapy (ACT) out of stock. Maps created by Toussaint Rouamba et al., 2019. Source of materials: The shapefile was obtained from the “Base Nationale de Découpage du territoire” of Burkina Faso (BNDT, 2006). The Service Availability and Readiness Assessment data for modelling were obtained from the Ministry of Health of Burkina Faso.

**Figure 4 ijerph-17-03923-f004:**
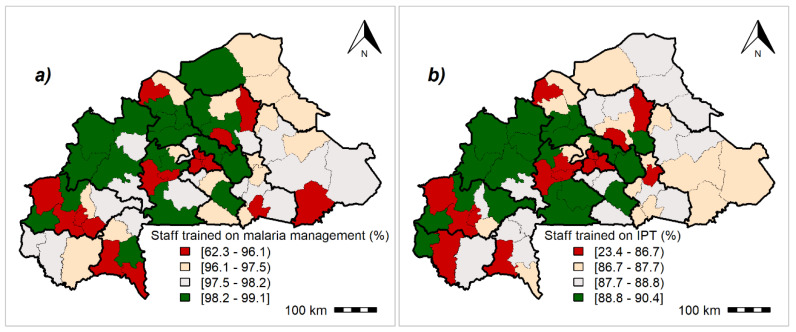
Geographical distribution of malaria readiness indicators regarding staff training at the health-district level: Posterior medians of fitted values. (**a**) Staff trained on guidelines for malaria diagnosis and treatment, (**b**) staff trained on guidelines for intermittent preventive treatment (IPT) of malaria during pregnancy. Maps created by Toussaint Rouamba et al., 2019. Source of materials: The shapefile was obtained from the “Base Nationale de Découpage du territoire” of Burkina Faso (BNDT, 2006). The Service Availability and Readiness Assessment data for modelling were obtained from the Ministry of Health of Burkina Faso.

**Figure 5 ijerph-17-03923-f005:**
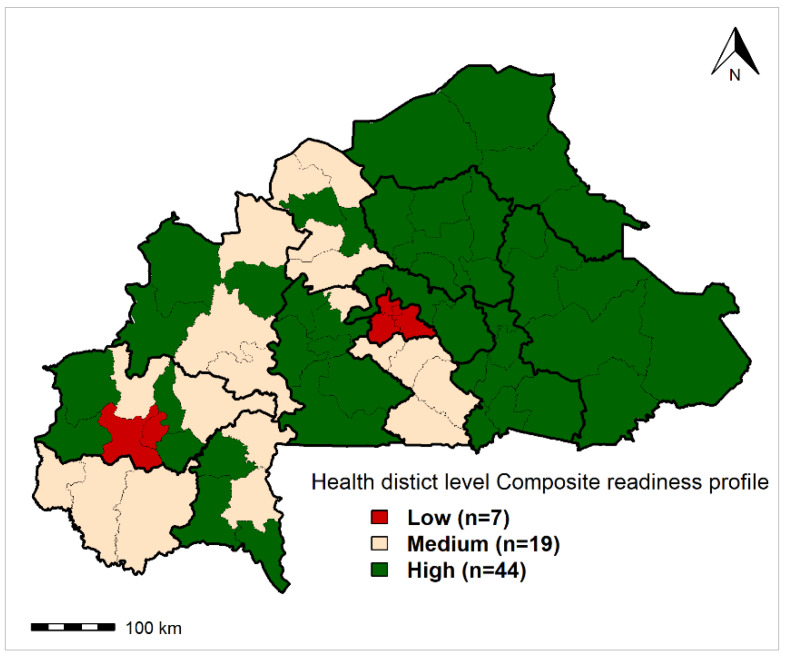
Geographical distribution of health district composite readiness profiles for malaria case management. The bold black lines represent the regional boundaries; the dashed black lines represent the health district boundaries. Maps created by Toussaint Rouamba et al., 2019. Source of materials: The shapefile was obtained from the “Base Nationale de Découpage du territoire” of Burkina Faso (BNDT, 2006). The Service Availability and Readiness Assessment data for modelling were obtained from the Ministry of Health of Burkina Faso.

**Table 1 ijerph-17-03923-t001:** Partition of sampled health facilities by health district, location, and governing authority.

Characteristic	Number of Health Districts	Health Facilities		
		Total (*N*)	Surveyed (*n*)	%
Total	70	2018	753	37.3
**Region**				
Boucle du Mouhoun	6	214	74	34.6
Cascades	3	84	29	34.5
Center	5	297	140	47.1
Center-east	7	136	51	37.5
Center-north	6	137	49	35.8
Center-west	7	194	68	35.1
Center-south	4	98	34	34.7
Est	6	135	49	36.3
Haut-Bassins	8	219	83	37.9
North	6	187	64	34.2
Plateau Central	3	127	44	34.6
Sahel	4	90	32	35.6
South-west	5	100	36	36.0
**Health-Facility Location**				
Rural			498	66.1
Urban			255	33.9
**Governing Authority**				
Public (Government)			596	79.2
NGO/Association/Private			116	15.4
Military			26	3.5
Confessional			15	2.0

Note: NGO: Nongovernmental organization; *N*: Total number of health faculties per region; *n*: number of surveyed health faculties.

**Table 2 ijerph-17-03923-t002:** Composite readiness profile characteristics of health districts obtained by hierarchical ascendant classification estimated from posterior means implemented in the Bayesian binomial model.

Readiness Indicator	Whole, *N* = 70	Health District Composite Readiness		
		Low (*n* = 7)	Medium (*n* = 19)	High (*n* = 44)
**Basic Equipment, % (IQR)**				
Adult weighing scale	95.7 (95.4–96.0)	96.4 (96.4–96.6)	95.7 (95.6–95.9)	95.7 (95.3–95.9)
Infant weighing scale	81.7 (65.8–87.3)	65.7 (59.5–66.7)	56.3 (53.1–62.1)	85.7 (81.9–90.5)
Stethoscope	98.9 (98.7–99.0)	96.6 (92.7–97.5)	98.9 (98.8–99.0)	98.9 (98.8–99.0)
Thermometer	99.8 (99.7–99.9)	99.2 (97.2–99.7)	99.8 (99.7–99.9)	99.9 (99.7–99.9)
Blood pressure apparatus	97.7 (97.3–98.0)	97.3 (96.4–97.8)	97.6 (97.2–97.9)	97.8 (97.4–98.1)
Light source	66.5 (58.7–79.6)	80.5 (76.3–82.1)	49.1 (32.4–62.5)	70.6 (63.4–81.3)
Latex gloves for physical exam	92.8 (88.5–96.4)	91.9 (85.9–93.5)	87.9 (80.1–92.2)	96.2 (90.9–97.9)
**Malaria Diagnosis, % (IQR)**				
Rapid diagnostic test availability	86.0 (75.9–89.8)	62.0 (55.7–65.4)	88.7 (81.6–90.8)	79.3 (66.4–87.4)
By clinical symptoms	98.9 (98.3–99.3)	97.9 (95.9–98.9)	98.8 (98.3–99.1)	99.1 (98.4–99.4)
By rapid diagnostic test	94.6 (94.2–95.0)	83.3 (68.8–84.4)	94.6 (94.3–95.0)	94.9 (94.6–95.1)
By microscopy	15.7 (11.9–20.0)	38.5 (35.0–47.4)	13.2 (11.8–16.3)	15.8 (11.7–18.7)
**Antimalaria Drugs, % (IQR)**				
First-line antimalarial (ACT)	96.5 (94.3–98.1)	51.2 (44.7–62.6)	96.5 (94.6–98.1)	97.9 (96.3–98.3)
Artesunate rectal or injectable forms	41.2 (28.1–46.5)	17.1 (12.7–22.6)	41.1 (28.8–46.6)	42.4 (36.6–48.4)
IPTp	93.3 (91.2–94.9)	71.0 (58.0–73.6)	93.2 (91.7–95.3)	94.6 (92.8–94.9)
ACT out of stock	04.6 (04.0–05.3)	05.9 (05.2–06.7)	04.6 (04.4–05.2)	04.4 (03.9–05.2)
**Staff Trained (and guidelines) % (IQR)**				
For diagnosis and treatment of malaria	97.5 (96.1–98.2)	77.4 (68.0–82.9)	97.5 (96.6–98.2)	98.0 (97.2–98.2)
For IPTp	87.7 (86.7–88.8)	54.2 (26.6–57.3)	87.7 (86.9–88.8)	88.6 (87.7–89.1)

Note: IQR: Interquartile range; ACT: Artemisinin-based combination therapy; IPTp: Intermittent preventive treatment during pregnancy; *N*: Total number of health faculties; *n*: Number of health faculties in each Composite Readiness.

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
