# Peer review of "How to Estimate Optimal Malaria Readiness Indicators at Health-District Level: Findings from the Burkina Faso Service Availability and Readiness Assessment (SARA) Data"

_ijerph, 2020, doi:10.3390/ijerph17113923_

Round 1
Reviewer 1 Report
The paper reports on the development of a model that can use locally generated data about malaria treatment and prevention preparedness to assess readiness at a regional level. The paper is very well written and the problem that is addressed is legitimate. There are a few changes recommended below:
Title line 2: Change "optimally" to "optimal"
Title line3: Change "?" to a colon (:)
Line 24: changed "causes to "contributors". The cause of death is malaria.
Line 33: Mandated by whom?
Line 54: "devastating" or "disproportionate. The disease also affects men in a devastating way; the issue and context are the disproportionate effect and pregnant women and children.
Line 57: insert "case fatality rates "
Line 130: Remove colon.
There is only one weakness that I see in this paper and it is common to papers on models. The authors have shown that the model can be generated but the degree to which the model actually reflects the real situation is not as evident. I would suggest a follow-up paper in validating the model, perhaps in another neighboring country or region. This does not prevent the publication of this manuscript, but perhaps this could be added as a recommendation for future research.
Nice manuscript.
Author Response
Minor
We would like to thank the reviewer for his valuable comments and suggestions. We appreciate these positive comments and the reviewers' efforts in reading our manuscript.
The paper reports on the development of a model that can use locally generated data about malaria treatment and prevention preparedness to assess readiness at a regional level. The paper is very well written and the problem that is addressed is legitimate.
There are a few changes recommended below:
Point 1:
Reviewer: Title line 2: Change "optimally" to "optimal"
Authors: Many thanks to the reviewer for making this suggestion. This is taking into account by changing the word “optimally” to “optimal”.
Please, see line 2 on the updated version of the manuscript.
Point 2:
Reviewer: Title line 3: Change "?" to a colon (:)
Authors: Many thanks to the reviewer for this suggestion. This is considering by reforming the nature of the sentence through the changing "?" to “:”
Please, see line 3 on the updated version of the manuscript.
Point 3:
Reviewer: Line 24: changed "causes to "contributors". The cause of death is malaria.
Authors: Many thanks to the reviewer for this remark. In the current version of the updated manuscript, we changed "causes to "contributors".
Please, see line 25 on the updated version of the manuscript.
Point 4:
Reviewer: Line 33: Mandated by whom?
Authors: Many thanks for raising this incomplete sentence. We have rephrased the sentence by specifying that the surveyed health-facilities were mandated by the Ministry of health to manage malaria cases.
Please, see this updated sentence on line 35 on the updated version of the manuscript.
Point 5:
Reviewer: Line 54: "devastating" or "disproportionate. The disease also affects men in a devastating way; the issue and context are the disproportionate effect and pregnant women and children.
Authors: We thank the reviewer for this suggestion. We have rephrased the sentence by replacing the word "devastating" by "disproportionate".
Please, see this updated sentence on line 55 on the updated version of the manuscript.
Point 6:
Reviewer: Line 57: insert "case fatality rates "
Authors: We thank the reviewer for this suggestion. We have rephrased the sentence by inserting the word "rates".
Please, see this updated sentence on line 58 on the updated version of the manuscript.
Point 7:
Reviewer: Line 130: Remove colon.
Authors: The colon is removed on the current revised version of the manuscript. Please, see line 133. Thanks.
Point 8:
Reviewer: There is only one weakness that I see in this paper and it is common to papers on models. The authors have shown that the model can be generated but the degree to which the model actually reflects the real situation is not as evident. I would suggest a follow-up paper in validating the model, perhaps in another neighboring country or region. This does not prevent the publication of this manuscript, but perhaps this could be added as a recommendation for future research.
Authors: We thank them very much for their valuable contribution. We raised this as a limitation, and suggested a study nested within the SARA surveys for validation of our models.
The formulation of the sentence is on line 398 – 402. “Another limitation is the fact that estimation of our indicators at the sub-national level are based on statistical models, which may not really reflect the real situation. For the upcoming SARA survey, we suggest a nested study where a region would be selected, in which a representative sample at the health district level will be performed to serve as a control district for the external validation of our model.”
Nice manuscript.
Reviewer 2 Report
The study estimates health district-level malaria readiness indicators from survey data. The authors use a binomial-hierarchical Bayesian spatial prediction approach applied to Burkina Faso Service Availability and Readiness Assessment (SARA) survey data.
This allows for estimates of essential equipment availability and readiness for malaria care at local scale. The analysis is technically correct and the manuscript is well written.
I have no specific comments.
Author Response
Point 1:
Reviewer: The study estimates health district-level malaria readiness indicators from survey data. The authors use a binomial-hierarchical Bayesian spatial prediction approach applied to Burkina Faso Service Availability and Readiness Assessment (SARA) survey data.
This allows for estimates of essential equipment availability and readiness for malaria care at local scale. The analysis is technically correct and the manuscript is well written.
I have no specific comments.
Authors: We would like to thank the reviewer for his valuable comments. We appreciate these positive comments and the reviewers' efforts in reading our manuscript.
Reviewer 3 Report
IJERPH_review May 15, 2020
Given that more than half of all malaria cases and deaths are in BF, this article appears to be an important contribution to the current literature in applied research.
Of note:
All figure legends need to be modified. When reviewing it is challenging (impossible) to differentiate given all is in greyscale. Additionally, anywhere script is present in the image, ideally it is a high resolution.
Line 141: could you explain how the verification and validation was done.
Line 238: should not be centered.
Figure 2. specifically, it is at first glance difficult to understand the rationale for the sequence, i.e. diagnosis could be in a triad of figures. Then the treatment is a quadrant of 4 panels. Staff training panels would be better if next to each other. Consider revising for better flow of information to readership. ACT is abbreviated in the figure but not in the figure legend.
Table 2: were any statistical analysis run for the comparative pairwise analysis. Which were statistically different? Could those be bolded.
Discussion: given the evaluation is done every 2 years, and given the variability of results that were found, might it be encouraged to consider that SARA be done more frequently (every 6 mths until a threshold had been reached, then every year after that?). For strong performing sites, they may remain at every 2 years. Perhaps a triaging system.
The limitations section is sparse. Were there no other limitations to the work to discuss…in the conclusion you list the limitations of the sites…
Perhaps it should be outlined in more detail who benefits from this work, is this article targeted for researchers, clinicians, public health officials, policy makers, health department of the government, primary/tertiary care physicians, medics, general population, all of the above.
References appear to be numbered twice. Please correct.
Author Response
Major
We would like to thank the reviewer for his valuable comments and suggestions. We appreciate these positive comments and the reviewers' efforts in reading our manuscript.
Given that more than half of all malaria cases and deaths are in BF, this article appears to be an important contribution to the current literature in applied research.
Of note:
Point 1:
Reviewer: All figure legends need to be modified. When reviewing it is challenging (impossible) to differentiate given all is in greyscale. Additionally, anywhere script is present in the image, ideally it is a high resolution.
Authors: We thank very much the reviewer for his valuable remark. The figure legends are modified to make readership to easily distinguish the different classes. Moreover, we increase the resolution of the graphs (ppi:300) to have high resolution.
Point 2:
Authors: We thank the reviewer very much. For the verification and internal validation of the model, we processed as follow: First, we executed the different models without the adjustment variables and second with the adjustment variables. And then, checked if there was an improvement on the value of the deviance information criterion (DIC) after the inclusion of the adjustment variables. The predictive performance (internal validation) of the models was assessed using cross-validation with the conditional predictive ordinate (CPO) approach (Pettit L, 1990). The convergence of model parameters was assessed graphically.
As suggested by a reviewer, we raised the external verification of the model as a limitation, and therefore make a perspective. For the upcoming SARA survey, we suggest a nested study where a region would be selected, in which a representative sample at the health district level will be performed to serve as a control district for the external validation of our model.
The explanation of the verification and validation was is on line 398 – 402 on the revised version of the manuscript.
Point 3:
Reviewer: Line 238: should not be centered.
Authors: Many thanks for this remark. This is resolved in the revised version of the manuscript. See line 253.
Point 4:
Reviewer: Figure 2. specifically, it is at first glance difficult to understand the rationale for the sequence, i.e. diagnosis could be in a triad of figures. Then the treatment is a quadrant of 4 panels. Staff training panels would be better if next to each other. Consider revising for better flow of information to readership. ACT is abbreviated in the figure but not in the figure legend.
Authors : We have revised and reorganized the different all the figures of the manuscript to allow better flow of information to readership. The acronym ACT is spelled out in the figure legend.
Point 5:
Reviewer: Table 2: were any statistical analysis run for the comparative pairwise analysis. Which were statistically different? Could those be bolded.
Authors: Yes, statistical analysis were running for the comparative pairwise analysis. We give a supplementary material to highlight variables that best characterize each composite readiness profile. For the SARA survey indicator variables (see Table S3), there is the mean of the variable in the cluster (Mean in the Profile), the mean of the variable for the data set (Overall Mean), the associated standard deviations and the p-value corresponding to the test of the following hypothesis: "the cluster mean is equal to the overall mean". A v.test value greater than 1.96 corresponds to a p-value less than 0.05; the sign of the v.test indicates whether the category mean is less or greater than the overall mean.
Point 6:
Reviewer: Discussion: given the evaluation is done every 2 years, and given the variability of results that were found, might it be encouraged to consider that SARA be done more frequently (every 6 mths until a threshold had been reached, then every year after that?). For strong performing sites, they may remain at every 2 years. Perhaps a triaging system.
Authors: Many thanks for this contribution. This suggestion is considered in the discussion. The following sentence is added (see lines 392 to 396):
“Given that in Burkina Faso context the evaluation of SARA indicators is assessed every two years, and given that our results found that there is a variability between health-districts it might be interesting that health authorities increased the frequency of SARA surveys especially for health district with weak performing. For instance, every six months until a desired threshold had been reached and every one after the threshold is reached, and then 2 years”
Point 7:
Reviewer: The limitations section is sparse. Were there no other limitations to the work to discuss…in the conclusion you list the limitations of the sites…
Authors: We have improved the limitation section by adding one weakness is common to papers on models. This weakness is formulated as follow ‘see line (398 to 402):
“Another limitation is the fact that estimation of our indicators at the sub-national level are based on statistical models, which may not really reflect the real situation. For the upcoming SARA survey, we suggest a nested study where a region would be selected, in which a representative sample at the health district level will be performed to serve as a control district for the validation of our model”
Point 8:
Reviewer: Perhaps it should be outlined in more detail who benefits from this work, is this article targeted for researchers, clinicians, public health officials, policy makers, health department of the government, primary/tertiary care physicians, medics, general population, all of the above.
Authors: detail of who benefits from this work is outlined in the manuscript. The following sentence is given in the revised version of the manuscript (see lines 248 to 261). “The results of our modeling are valuable for informing malaria control program managers, technical and financial partners, and Ministry of Health officials at the health district, provincial, regional, and central levels. The findings regarding the tracers of the readiness for services in the management of malaria indicated deficiencies that vary between health districts and need to be addressed by all stakeholders”.
Point 9:
Reviewer: References appear to be numbered twice. Please correct.
Authors: The references are checked and corrected
Round 2
Reviewer 3 Report
None.